# Exclusion of enrolled participants in randomised controlled trials: what to do with ineligible participants?

Andrea M Rehman ,[1] Rashida Ferrand,[2,3] Elizabeth Allen,[4] Victoria Simms,[1,3] Grace McHugh,[3] Helen Anne Weiss [1]

[1]MRC International Statistics & Epidemiology Group, London School of Hygiene and Tropical Medicine, London, UK
[2]Department of Clinical Research, London School of Hygiene and Tropical Medicine, London, UK
[3]Biomedical Research and Training Institute, Harare, Zimbabwe
[4]Department of Medical Statistics, London School of Hygiene and Tropical Medicine, London, UK

**Correspondence to**
Dr Andrea M Rehman;
andrea.rehman@lshtm.ac.uk

## ABSTRACT

**Objective** Post-randomisation exclusions in randomised controlled trials are common and may include participants identified as not meeting trial eligibility criteria after randomisation. We report how a decision might be reached and reported on, to include or exclude these participants. We illustrate using a motivating scenario from the BREATHE trial (Trial registration ClinicalTrials.gov, NCT02426112) evaluating azithromycin for the treatment of chronic lung disease in people aged 6–19 years with HIV in Zimbabwe and Malawi.

**Key points** Including all enrolled and randomised participants in the primary analysis of a trial ensures an unbiased estimate of the intervention effect using intention-to-treat principles, and minimises the effects of confounding through balanced allocation to trial arm. Ineligible participants are sometimes enrolled, due to measurement or human error. Of 347 participants enrolled into the BREATHE trial, 11 (3.2%) were subsequently found to be ineligible based on lung function criteria. We assumed no safety risk of azithromycin treatment; their inclusion in the trial and subsequent analysis of the intervention effect therefore mirrors clinical practice. Senior trial investigators considered diurnal variations in the measurement of lung function, advantages of retaining a higher sample size and advice from the Data Safety and Monitoring Board and Trial Steering Committee, and decided to include these participants in primary analysis. We planned and reported analyses including and excluding these participants, and in our case the interpretation of treatment effect was consistent.

**Conclusion** The decision, by senior investigators, on whether to exclude enrolled participants, should reflect issues of safety, treatment efficacy, statistical power and measurement error. As long as decisions are made prior to finalising the statistical analysis plan for the trial, the risk of exclusions creating bias should be minimal. The decision taken should be transparently reported and a sensitivity analysis can present the opposite decision.

## MOTIVATING EXAMPLE

In an individually randomised placebo-controlled trial (registered ClinicalTrials.gov, NCT02426112) of the impact of azithromycin on treatment of chronic lung disease in children and adolescents born with HIV in Zimbabwe and Malawi, one eligibility criterion was a measurement cut-off for lung function (using forced expiratory volume in 1 second ($FEV_1$ z-score).[1 2] After enrolment was complete, and prior to data analysis, inconsistencies were identified with the $FEV_1$ inclusion criteria. Specifically, height (an input variable for the reference equations on the European Respiratory Society/Global Lung function Initiative 2012 spreadsheet to compute the $FEV_1$ z-score,[3]) measured at screening did not always align with height measurements from later study visits. A review of practices undertaken found that in one country, different models of stadiometers were used at different screening centres resulting in inconsistencies in height measurements. It was decided to recalculate the z-score using a mean of height from two later study visits in that country, and in the other country to use a mean height from screening and two later study visits (up to 2 weeks after randomisation). These recalculations meant that 11/347 (3.2%) participants fell outside the lung function cut-off for inclusion into the trial and a debate ensued among the trial investigators as to how to proceed. The first stage was to unmask these participants to the local study physician, report this protocol violation to the Data Safety and Monitoring Board (DSMB) and the Trial Steering Committee (TSC) and to the relevant ethics committee(s).

At this stage, 7 of the 11 ineligible participants had completed their course of study medication and the remaining four participants had between 1 and 3 weeks of study medication remaining. The trial drug, azithromycin, is considered safe,[4] so the potential for harm in continuing the four participants on treatment was considered low. The initial suggestion from the investigators was to withdraw these participants from the trial and stop treatment, but the DSMB and TSC advised that they should be included as the primary analysis, with a sensitivity analysis

excluding them. The reason for this was (1) lung function may vary by time of day,[5] so if lung function had been retested at a different time even on the same day, eligibility may have been different, (2) the low risk of treatment-related adverse events and (3) the advantage in retaining a greater sample size (original power calculations required 300 for primary outcome at 12 months). Further, a prespecified subgroup analysis was included in the statistical analysis plan to investigate effect modification by baseline $FEV_1$ measurement, and provide estimates of treatment effect at different severities of baseline lung function. We also considered the possible adverse impact on statistical power if ineligible participants were less responsive to azithromycin treatment than eligible participants (potentially resulting in greater variability of treatment effect). On balance, this was outweighed by reasons to include ineligible participants.

## STATISTICAL CONSIDERATIONS

The intention-to-treat (ITT) principle underpins the analysis of randomised controlled trials as a means of obtaining balance between arms on potential confounding factors, and preventing estimates of the intervention effect from being biased.[6 7] ITT in its purest form estimate the intervention effect for all enrolled randomised participants, based on the initial arm allocation. In practice, the trial outcomes may not be measured or analysed for all randomised participants and this may impact on the ITT principle. One reason for not measuring outcomes in all enrolled participants is attrition, causing missing outcome data. If outcome data are missing not at random the ITT principle can become compromised.[8] A second reason, which compromises the ITT principle, but is common nonetheless,[9] is that some randomised participants may

| Table 1 | Reasons for and against post-randomisation exclusions | |
|---|---|---|
| **Issue** | **Reason to include** | **Reason to exclude** |
| *Clinical scenario* | | |
| Make recommendations of benefit or harm (based on trial results) relating to a certain patient population | Where there is uncertainty over defining patient populations, it would be a conservative approach to retain all participants. | Retains a defined group of included participants meeting inclusion/exclusion criteria neatly in which the intervention is hypothesised to be the most effective. |
| Disease status may be unclear | Measurement cut-off may not relate to a 'disease' state and may be arbitrary. | Measurement cut-offs are commonly used to indicate disease severity although knowing there may be some misclassification. |
| Assessment of safety risks | There is no safety risk to participants after review and therefore treatment and follow-up can continue. | Randomisation was mistakenly done, for example when found not to be diseased. Where safety was compromised the participants should cease remaining treatment and most likely be excluded from analysis. |
| *Statistical analysis* | | |
| Maintain ITT principles, providing an unbiased treatment effect | Stays true to ITT principle ensuring balance on known and unknown factors between arms when all enrolled and randomised participants are analysed. | The risk of bias from excluding some participants has been shown to be low under certain conditions. |
| The inclusion criteria are subject to measurement error. The relationship between the inclusion criteria and the primary outcome should be considered. | Pragmatically, errors in measurement will occur in routine practice. They may have been considered eligible at the point of enrolment. Include if measurement of the primary outcome is not impacted by measurement error in the inclusion criteria. | Identification of errors in the measurement of disease state and excluding them can prevent underestimation of treatment effects. |
| Effect on statistical power | A larger sample size is retained. | If ineligible participants' responses to treatment differ from those for eligible participants (eg, less response), the variance of the primary outcome may be increased meaning there may be more statistical power to exclude them. |
| *Integrity and transparency* | | |
| Justifying the decision to include or exclude | Post-randomisation exclusions may be mistrusted in the scientific community if conflicts of interest or the trial sponsor are shown to have influenced the decision-making. | Post-randomisation exclusions are a common approach in the scientific community and will be accepted when clearly justified. |

*ITT intention-to-treat

**Table 2** Baseline characteristics of BREATHE trial participants stratified by eligibility for inclusion

| Characteristic | Eligible randomised participants n=336 | Eligible analysed participants n=297 | Ineligible randomised and analysed participants n=11* |
|---|---|---|---|
| Placebo arm, n (%) | 170 (51) | 142 (48) | 4 (36) |
| AZM arm, n (%) | 166 (49) | 155 (52) | 7 (64) |
| Baseline FEV$_1$ z-score, mean (SD) | −2.05 (0.72) | −2.05 (0.73) | −0.67 (0.38) |
| 48-week FEV$_1$ z-score, mean (SD) | − | −1.95 (0.90) | −1.24 (0.84) |
| Zimbabwe site, n (%) | 241 (72) | 219 (74) | 0 (0) |
| Malawi site, n (%) | 95 (28) | 78 (26) | 11 (100) |
| Aged 6–10, n (%) | 44 (13) | 40 (13) | 3 (27) |
| Aged 11–15, n (%) | 152 (45) | 135 (45) | 6 (55) |
| Aged 16–19, n (%) | 140 (42) | 122 (41) | 2 (18) |
| Female sex, n (%) | 166 (49) | 142 (48) | 4 (36) |
| Male sex, n (%) | 170 (51) | 155 (52) | 7 (64) |
| Baseline log10 HIV viral load, mean (SD)† | 2.79 (1.61) | 2.72 (1.59) | 2.32 (1.95) |
| Baseline suppressed HIV viral load (<1000 copies/mL), n (%)† | 187 (56) | 171 (58) | 7 (64) |

*All ineligible randomised participants were analysed for the primary outcome.
†N=2 missing values among eligible participants were imputed in the primary analysis using chained equations.
AZM, azithromycin; FEV$_1$, forced expiratory volume in 1 second.

be excluded from analysis post-randomisation. Reasons for such exclusions might be that participants (1) have incomplete baseline or outcome data, (2) did not receive the intervention allocated or (3) were found to be ineligible post-randomisation. In this communication, we summarise issues to consider when deciding whether to exclude enrolled, but ineligible, participants during the analysis of the intervention effect (table 1).

There is conflicting evidence as to whether post-randomisation exclusions of enrolled participants produce bias.[10–13] Bias can be considered a potential issue where decisions about exclusions are influenced by the trial sponsor or conflicts of interest of the investigators.[14]

The statistical power of the trial may be affected in either direction when ineligible individuals are enrolled incorrectly. Including the ineligible enrolled participants in analysis will retain a larger sample size, while excluding them may increase the variance of the estimated intervention effects (if those ineligible were to respond differently to treatment than eligible participants).

The type of inclusion/exclusion criteria must be considered. For example, in a drug treatment trial providing treatment for a certain infection, if it was found post-randomisation that an enrolled participant was uninfected it is best to exclude that participant from analysis and withdraw them immediately from the trial. Decisions are less clear if (1) the criteria include a cut-off used for inclusion (eg, body mass index), and there is error in the measurement of this, or (2) exclusion criteria include the presence of a clinical condition for which screening tests were not available at enrolment and only become apparent during follow-up.

Depending how quickly it became apparent that a participant did not meet eligibility criteria, trial outcome data may have been collected on ineligible enrolled participants and a decision must be made whether to include them in primary analysis. If excluded, a 'modified ITT' may be performed.[15–17]

## REPORTING AND REFLECTION ON MOTIVATING EXAMPLE

The primary outcome was analysed for 308 participants, 11 of whom were ineligible based on FEV$_1$ inclusion criteria. By chance, differences were observed between trial arms in age and sex distributions and with HIV-related characteristics. Primary analyses were therefore prespecified, prior to unmasking of outcome data, to adjust for site, age, sex and HIV viral load. Once-weekly administration of azithromycin did not improve lung function measured by FEV$_1$ z-score after 48 weeks in ITT analysis (adjusted mean difference (aMD) 0.06%, 95% CI −0.10% to 0.21%) and in sensitivity analysis excluding those who did not meet eligibility criteria (aMD 0.07%, 95% CI −0.08% to 0.23%).[2] The prespecified per-protocol analysis suggested weak evidence for an effect of azithromycin, with an aMD in z-scores of 0.14 (95% CI −0.02 to 0.29) favouring azithromycin. Those not meeting eligibility criteria were more likely to be in the azithromycin arm, in Malawi, younger, of male sex and have HIV viral suppression (table 2).

The study was powered to detect a 0.32 z-score difference between trial arms with 300 participants assuming a mean z-score of −2.04 (SD 0.82) in the placebo arm. The primary outcome was assessed in 308 participants, with a mean of −1.95 (SD 0.91) in the placebo arm. Effectively,

with a sample size of 146 in the placebo arm and 162 in the azithromycin arm, the study had 80% power to detect a 0.29 z-score difference between trial arms; excluding ineligible participants gave the same z-score difference.

In practice, the inclusion of ineligible participants did not change the interpretation of the trial results, likely due to their low numbers and/or because the adjustments used for primary analysis (to account for baseline imbalance) were also associated with ineligibility (and being assessed for the primary outcome). The study remained sufficiently powered. Sensitivity analyses were prespecified in a formal statistical analysis plan, shared with reviewers and reported in the publication of the trial findings for transparency and to maintain research integrity.

## CONCLUSION

There is not a one-size-fits-all approach to deciding on post-randomisation exclusions and in fact, there is evidence to suggest that more trials tend to make post-randomisation exclusions than do not.[9] Consideration should be given to safety, assessment of treatment effects, statistical power and measurement error (table 1). We recommend that the decision is made after a joint discussion among senior trial investigators in conjunction with the TSC and DSMB. Others may advise, but the final decision falls to the senior investigators of the trial who should not be influenced by the trial sponsor or conflicts of interest, financial or otherwise. To further reduce bias, a decision should be made *prior* to finalising the statistical analysis plan for the trial, and for transparency, reported explicitly when publishing the trial results. Justification for including or excluding the participants who were found not to meet inclusion criteria after randomisation should be presented for scrutiny by the scientific community and it may be appropriate to consider a sensitivity analysis using the opposite decision. The aim of any decision is to remain as close to ITT principles as possible and present an unbiased estimate of the treatment effect.

**Contributors** AMR prepared the first draft of manuscript. AMR, RF, EA, VS, GM and HAW were involved in the preparation and conception of the manuscript, critically reviewed the manuscript and approved the submitted version.

**Funding** The BREATHE trial was funded by the Global Health and Vaccination Research (GLOBVAC) Programme of the Medical Research Council of Norway. RF is funded by the Wellcome Trust (Grant Number 206316/Z/17/Z). HAW, AMR and VS were supported in part by a grants from the Medical Research Council (MRC) and the Department for International Development (DFID UK) under the MRC/DFID Concordat (K012126/1) and is also part of the EDCTP2 programme supported by the European Union (Grant Number MR/R010161/1).

**Competing interests** None declared.

**Patient consent for publication** Not required.

**Provenance and peer review** Not commissioned; externally peer reviewed.

**Open access** This is an open access article distributed in accordance with the Creative Commons Attribution 4.0 Unported (CC BY 4.0) license, which permits others to copy, redistribute, remix, transform and build upon this work for any purpose, provided the original work is properly cited, a link to the licence is given, and indication of whether changes were made. See: https://creativecommons.org/licenses/by/4.0/.

**ORCID iDs**
Andrea M Rehman http://orcid.org/0000-0001-9967-5822
Helen Anne Weiss http://orcid.org/0000-0003-3547-7936

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
