## [Reviewer comments · BMJ Open]

ARTICLE DETAILS

TITLE (PROVISIONAL)	Exclusion of enrolled participants in randomised controlled trials: what to do with ineligible participants?
AUTHORS	Rehman, Andrea; Ferrand, Rashida; Allen, Elizabeth; Simms, Victoria; McHugh, Grace; Weiss, Helen

VERSION 1 – REVIEW

REVIEWER	Amir Sarayani University of Florida, USA
REVIEW RETURNED	07-Sep-2020

GENERAL COMMENTS	Thank you for the opportunity to review this manuscript. The authors have communicated their experience with mismeasurement in assessing the eligibility criteria for a randomized controlled trial (BREATRHE), and their thought process to address the issue post-randomization. I have a few comments that might enhance the message of this communication piece: 1. Table 1 summarises the issues that might arise in decision-making for this challenge. It might be helpful to organize the issues in subsection like issues related to the clinical scenario, statistical analysis, integrity & transparency2. There are a few instances that may need language edits: "for example having the disease under study in a in a drug treatment trial" "This occurs commonly,[4] and again comprises the ITT principle." "Consider whether measurement error in the inclusion criteria does not impact on the measurement of the primary outcome."3. Would the author consider a per-protocol analysis approach with appropriate causal inference methods as a potential solution or a sensitivity analysis when post-randomization exclusion happens?
---

REVIEWER	Simon Kolstoe University of Portsmouth Portsmouth UK
REVIEW RETURNED	10-Oct-2020

GENERAL COMMENTS	The authors experience with the BREATHE trial, and the unintentional inclusion of 11 participants who did not meet the study inclusion criteria, is an interesting case study that raises important
---

	issues. Generally I support the publication of this reflection, but suggest the article needs restructuring. At present the introduction, background and considerations sections do not flow particularly well, and only start to make sense to the reader once they have read the "Example" section. Following reading the example I then had to go back and re-read the earlier sections which now seemed far more relevant! At the moment the piece reads like a very brief, even incomplete, philosophical/ethical/methodological reflection followed by a really interesting example. Assuming the editors agree, I would suggest being open right from the start of this manuscript that you are describing a specific situation and subsequent actions/reflections that you then made. Put the description of the events that occurred in your study (that are currently in the "Example" section) right at the beginning to make up the "Background" section. This seems to make most sense logically as this experience was the background for your considerations as to "what to do with ineligible participants" - I imagine if you had not encountered this problem you would not be publishing this manuscript. Much of the text in the current background and considerations sections can then go in the discussion following your example. This way the reader can clearly see the relevance of your reflections to the situation that you encountered. Generally articles on these types of topics are either written by philosophers/ethicists who create a carefully reasoned argument before adding examples to support the argument, or conversely written by scientists/clinicians who have encountered an interesting situation and then reflected on it. Your article is clearly of the latter type, so must reflect this in its structure. It would also be helpful to add some further details about the original study - perhaps in the supplementary information or a table. I would be particularly interested to see a break-down of the participants and thus the unintentional inclusions within the context of the other participants. Some specific information as to how the inclusion/exclusion of the ineligible participants affected the power of the study (and different outcome measures) would also be helpful. If the changes suggested above were made, the abstract would finally also need re-writing to reflect the new structure of the article.
--	--

VERSION 1 – AUTHOR RESPONSE

Reviewer: 1
Reviewer Name

Amir Sarayani

Institution and Country

University of Florida, USA

Please state any competing interests or state 'None declared':
None

Thank you for the opportunity to review this manuscript. The authors have communicated their experience with mismeasurement in assessing the eligibility criteria for a randomized controlled trial (BREATHE), and their thought process to address the issue post-randomization.

I have a few comments that might enhance the message of this communication piece:

1. Table 1 summarises the issues that might arise in decision-making for this challenge. It might be helpful to organize the issues in subsection like issues related to the clinical scenario, statistical analysis, integrity & transparency

We thank the reviewer for this suggestion which improves the table.

2. There are a few instances that may need language edits:

"for example having the disease under study in a in a drug treatment trial"

"This occurs commonly,[4] and again comprises the ITT principle."

"Consider whether measurement error in the inclusion criteria does not impact on the measurement of the primary outcome."

We have made language edits to the highlighted sentences.

3. Would the author consider a per-protocol analysis approach with appropriate causal inference methods as a potential solution or a sensitivity analysis when post-randomization exclusion happens?

We carried out the latter and now include the details comparing the primary analysis and sensitivity analysis results from the BREATHE study which was accepted for publication in JAMA Network Open in October 2020.

Reviewer: 2

Reviewer Name

Simon Kolstoe

Institution and Country

University of Portsmouth

Portsmouth

UK

Please state any competing interests or state 'None declared':

None declared

The authors experience with the BREATHE trial, and the unintentional inclusion of 11 participants who did not meet the study inclusion criteria, is an interesting case study that raises important issues. Generally I support the publication of this reflection, but suggest the article needs restructuring.

At present the introduction, background and considerations sections do not flow particularly well, and only start to make sense to the reader once they have read the "Example" section. Following reading the example I then had to go back and re-read the earlier sections which now seemed far more relevant! At the moment the piece reads like a very brief, even incomplete, philosophical/ethical/methodological reflection followed by a really interesting example.

Assuming the editors agree, I would suggest being open right from the start of this manuscript that you are describing a specific situation and subsequent actions/reflections that you then made. Put the description of the events that occurred in your study (that are currently in the "Example" section) right at the beginning to make up the "Background" section. This seems to make most sense logically as this experience was the background for your considerations as to "what to do with ineligible participants" - I imagine if you had not encountered this problem you would not be publishing this manuscript. Much of the text in the current background and considerations sections can then go in the discussion following your example. This way the reader can clearly see the relevance of your

reflections to the situation that you encountered.

Generally articles on these types of topics are either written by philosophers/ethicists who create a carefully reasoned argument before adding examples to support the argument, or conversely written by scientists/clinicians who have encountered an interesting situation and then reflected on it. Your article is clearly of the latter type, so must reflect this in its structure.

We thank the reviewer for their thoughtful evaluation of the structure of our manuscript and have amended the structure as advised.

It would also be helpful to add some further details about the original study - perhaps in the supplementary information or a table. I would be particularly interested to see a break-down of the participants and thus the unintentional inclusions within the context of the other participants. Some specific information as to how the inclusion/exclusion of the ineligible participants affected the power of the study (and different outcome measures) would also be helpful.

We have added this information into the revised manuscript.

If the changes suggested above were made, the abstract would finally also need re-writing to reflect the new structure of the article.

This has been reworked.